# Evaluation of Silk Fibroin/Gellan Gum Hydrogels with Controlled Molecular Weight through Silk Fibroin Hydrolysis for Tissue Engineering Application

**DOI:** 10.3390/molecules28135222

**Published:** 2023-07-05

**Authors:** Sunjae Park, Soo-In Kim, Joo-Hee Choi, Se-Eun Kim, Seung-Ho Choe, Youngjun Son, Tae-woong Kang, Jeong-Eun Song, Gilson Khang

**Affiliations:** 1Department of PolymerNano Science & Technology and Polymer Materials Fusion Research Center, Jeonbuk National University, 567 Baekje-daero, Deokjin-gu, Jeonju-si 54896, Jeonbuk, Republic of Korea; sunjaepark@jbnu.ac.kr (S.P.); 201614309@jbnu.ac.kr (S.-I.K.); zooheechoi@jbnu.ac.kr (J.-H.C.); sen@jbnu.ac.kr (S.-E.K.); tmdgh8898@jbnu.ac.kr (S.-H.C.); spring19963@jbnu.ac.kr (Y.S.); taewoongkang@jbnu.ac.kr (T.-w.K.); songje@jbnu.ac.kr (J.-E.S.); 2Department of Bionanotechnology and Bio-Convergence Engineering, Jeonbuk National University, 567 Baekje-daero, Deokjin-gu, Jeonju-si 54896, Jeonbuk, Republic of Korea; 3Department of Orthopaedic & Traumatology, Airlangga University, Jl. Airlangga No. 4–6, Airlangga, Kec. Gubeng, Kota SBY, Surabaya 60115, Jawa Timur, Indonesia

**Keywords:** silk fibroin, gellan gum, dual-crosslinked hydrogel, hydrolysis, tissue engineering

## Abstract

Hydrogel is a versatile material that can be manipulated to achieve the desired physicochemical properties, such as stiffness, pore size, and viscoelasticity. Traditionally, these properties have been controlled through parameters such as concentration and pH adjustments. In this study, we focused on exploring the potential of hydrolyzed silk fibroin (HSF) as a molecular weight-modulating agent to control the physicochemical properties of double-composite hydrogels. We developed a synergistic dual-crosslinked hydrogel by combining ionically crosslinked silk fibroin with gellan gum (GG). The hydrolysis of silk fibroin not only enhanced its hydrophilicity but also enabled adjustments in its mechanical properties, including the pore size, initial modulus elasticity, and relaxation time. Moreover, biocompatibility assessments based on cell viability tests confirmed the potential of these hydrogels as biocompatible materials. By highlighting the significance of developing an HSF/GG dual-crosslinked hydrogel, this study contributes to the advancement of novel double-composite hydrogels with remarkable biocompatibility. Overall, our findings demonstrate the capability of controlling the mechanical properties of hydrogels through molecular weight modulation via hydrolysis and highlight the development of a biocompatible HSF/GG dual-crosslinked hydrogel with potential biomedical applications.

## 1. Introduction

Hydrogel is a versatile material that can be engineered to have specific physicochemical properties, such as stiffness, pore size, viscoelasticity, microarchitecture, degradability, ligand presentation, and stimulus responsiveness [1]. By manipulating these properties, hydrogels can be tailored to modulate cell signaling pathways and cellular fate, making them suitable for various biomedical applications. Hydrogels have emerged as promising platforms for drug delivery due to their ability to encapsulate drugs and provide sustained and controlled release [2,3]. The three-dimensional polymeric network structure of hydrogels enables them to entrap drugs within their matrix, ensuring their stability and preventing premature degradation. This feature is crucial for maintaining drug efficacy and optimizing therapeutic outcomes. Moreover, hydrogels exhibit exceptional potential in tissue regeneration and disease therapy. By tailoring their physical and chemical properties, hydrogels can mimic the extracellular matrix, providing a supportive environment for cell growth, migration, and differentiation. This property makes them ideal candidates for applications in tissue engineering, where they can promote the regeneration of damaged tissues and organs.

In the pursuit of advanced hydrogel materials, by combining these materials, researchers aim to develop composite hydrogels that exhibit complementary characteristics, harnessing the benefits of each component while overcoming their drawbacks.

Silk fibroin (SF), a natural protein obtained from silk fibers of silkworms, has gained significant attention as a versatile biomaterial due to its exceptional properties and extensive applications in regenerative medicine and tissue engineering. SF exhibits biocompatibility, minimal inflammatory responses, and low immunogenicity when used as a biomaterial [4,5,6]. The presence of a β-sheet (crystalline)-rich structure in SF contributes to its remarkable mechanical properties, while its amphiphilic nature enables the fabrication of SF in various forms, such as hydrogels, membranes, and porous sponges [7,8,9]. The fabrication of silk-based hydrogels involves transforming the amorphous random coil conformation of silk into a crystalline β-sheet structure [10].

Gellan gum (GG) is becoming increasingly popular for biomedical applications due to its stability, desirable mechanical properties, resistance to acid and heat, and sensitivity to ions. It is an extracellular polysaccharide composed of repeating units of β-d-glucose, l-rhamnose, and d-glucuronic acid in a 2:1:1 molecular ratio [11,12]. The presence of cationic agents such as Na^+^, K^+^, Ca^2+^, and Mg^2+^ enables ionic crosslinking, eliminating the need for chemical crosslinking agents that may interfere with cell growth and differentiation [13]. However, GG itself lacks cell adhesion sites, creating an unfavorable microenvironment for cell growth [14].

The ionic crosslinking capability of gellan gum (GG) enables the creation of composite scaffolds that combine the advantages of multiple biomaterials [15,16]. Among these, the SF/GG hydrogel can be easily developed by crosslinking the overall matrix with calcium ions, while the SF trapped within the GG structure gradually forms β-sheets in an aqueous environment. However, when multiple biomaterials are crosslinked, controlling the mechanical properties of the scaffold becomes a complex task with several variables to consider. SF, in particular, exhibits different physical properties depending on the arrangement of its chains. Factors such as the concentration, salt/ions, pH, and sonication time influence SF chain alignment, which affects the mechanical properties and gelation behavior [17,18]. However, previous studies have shown that except for differences in SF solution concentrations, inducing significant changes in both the mechanical and physical properties has been challenging [19]. 

In this study, rather than considering other variables, we attempted to modify the physical properties of SF by controlling its molecular weight through hydrolysis (HSF). Additionally, SF contains hydrophobic residues such as alanine and glycine [20], which may hinder uniform mixing with GG. Through hydrolysis, the creation of new hydrophilic N- and C-termini occurs at the ends of the protein fragments. These newly formed termini can interact with surrounding water molecules, leading to a change in the overall hydrophobicity of the molecule. In terms of scaffolds, the hydrophilic property can promote cell proliferation by facilitating cell–cell and cell–scaffold interactions. A hydrophilic surface enables the exchange of growth factors, cytokines, and other signaling molecules between neighboring cells, promoting tissue formation and cell proliferation [21,22]. Therefore, we aimed to investigate how the physical properties of SF change through hydrolysis and how the increased hydrophilicity influences cell growth in this study.

## 2. Results and Discussion

### 2.1. Properties of Hydrolyzed Silk Fibroin

The hydrogel network structures were formed through the self-assembly of solubilized SF molecules, facilitated by physical crosslinking between hydrophobic segments [19]. To control the mechanical properties and gelation behavior, hydrolysis was performed by adding 3.5 mL of 1 M hydrochloric acid solution at specific time points, as described in the Section 3.

Hydrolysis of SF was assessed by performing SDS-PAGE, and the results are shown in Figure 1. SF macromolecules consist of three types of peptide chains: light chain (25 kDa), heavy chain (325 kDa), and P25 glycoprotein (26 kDa) [23]. Due to the SF degumming process, the prepared protein primarily exhibited molecular weights concentrated at 200 kDa or higher. However, it was observed that the bands on the SDS-PAGE gel decreased in intensity as the hydrolysis time increased. This observation indicates that the bonds between SF chains can be efficiently broken through hydrolysis.

### 2.2. Physicochemical Properties of HSF/GG Hydrogel

HSF/GG hydrogels were fabricated through ionic crosslinking facilitated by the addition of Ca^2+^. Initially, hydrogels were prepared using 0.5% GG, 1.5% SF, and 0.03% CaCl_2_, with the only variable being the hydrolysis time. 

The FT-IR spectra of HSF and HSF/GG hydrogels are shown in Figure 2A in the range of 500–4000 cm^−1^. GG-specific peaks were detected at 3300 cm^−1^ (OH stretching group), 1596 cm^−1^ (CH_2_ stretching vibration group), and 1029 cm^−1^ (COC stretching group), respectively [24]. Three characteristic peaks can be seen in the pure SF at 3300 cm^−1^ (NH stretching group), 1636 cm^−1^ (amide I), 1556 cm^−1^ (amide II), and 1232 cm^−1^ (amide III) [25]. As shown in Figure 2A, the peak values in the composite hydrogel were 3385 cm^−1^ (OH and NH stretching group), 1643 cm^−1^ (amide I), 1515 cm^−1^ (amide II), 1234 cm^−1^ (amide III), and 1029 cm^−1^ (COC stretching). There was no significant difference observed in the peak values between pure GG and SF, as mentioned earlier. However, a slight peak shift was observed, which can be attributed to the alteration in the molecular structure induced by the ion crosslinking process [26]. It was confirmed that the presence of calcium ions resulted in physical interactions rather than the formation of new peaks.

The morphological characteristics of the hydrogels were examined using scanning electron microscopy (SEM), as illustrated in Figure 2B. The control and manipulation of the pore size in hydrogels are crucial for tailoring their properties to suit various applications. As depicted in Figure 2B, it was observed that the pore size could be regulated by adjusting the molecular weight of SF through hydrolysis. With an increased hydrolysis time, the amount of fragmented SF increased, leading to the formation of smaller SF clusters. Notably, the HSF180/GG experimental group exhibited the largest average pore size (130 ± 10 μm), indicating that the formation of smaller SF clusters contributed to the development of a larger pore network.

Just as the molecular weight adjusted by hydrolysis affected the pore size, it was also confirmed how it affected the physical and chemical properties, such as mass swelling, the sol fraction, and the degradation ratio. Generally, the sol fraction represents the initial mass of the hydrogel that can dissolve or disperse in a solvent. Therefore, it can be said that the lower the sol fraction value means the higher degree of crosslinking and the more stable hydrogel structure [27]. Here, HSF0/GG, HSF60/GG, and HSF180/GG showed sol fractions of 50 ± 5%, 56 ± 3%, and 75 ± 2%, respectively (Figure 3A). Typically, hydrogels with a higher sol fraction tend to exhibit higher mass swelling ratios [28]. As revealed in Figure 3B, it was confirmed that hydrogels exhibiting a higher sol fraction exhibited a higher mass swelling ratio. The reason for this is thought to be that the hydrophilicity of the SF segments is increased by removing hydrophobic amino acid residues such as alanine and glycine through hydrolysis [29]. 

From the viewpoint of drug delivery or biocompatibility, the degradation rate of hydrogel is also a factor to be considered. If degradation is too rapid, drug delivery to the target site may not be effective. Conversely, if it decomposes too slowly, it may interfere with being replaced by its own tissue, so it must have appropriate degradation characteristics [30]. SF hydrogels degrade slowly over several weeks to months. In this study, the degradation ratio was observed up to 28 days (Figure 3C), but it was confirmed that HSF0/GG, HSF60/GG, and HSF180/GG were slowly degraded, with weight ratios of 6.45 ± 0.94%, 6.41 ± 1.02%, and 6.61 ± 0.86%, respectively. It was confirmed that SF fragmented due to hydrolysis did not have a significant effect on the degradation ratio [31].

### 2.3. Mechanical Properties of HSF/GG Hydrogel

The compressive strength, viscosity, and injectability of hydrogels are important considerations as they need to maintain their inherent physical properties against external forces until the regenerated area is fully healed. As depicted in Figure 4A, the viscosity of the hydrogel was evaluated in response to temperature changes. Gelation occurred at approximately 31.3 °C in all experimental groups. During the process of polymer gelation, it is crucial to encapsulate cells under conditions similar to physiological temperatures, ensuring the formation of a uniform concentration gradient of cells without negatively impacting cell growth. Our findings indicated that, within the range investigated, the gelation temperature of the hydrogel did not significantly affect cell proliferation in vitro.

In order to determine whether the produced hydrogel was suitable for injection, the injectability was measured, as shown Figure 4B. The ability of the hydrogel to transport certain types of cargo that are sensitive to mechanical forces, such as cells, was also affected by its injectability. While many factors such as flow rate, needle gauge, and needle length can influence the injectability, the clinically accepted injection force is below 20 N [32]. As all hydrogels exhibited a measured force of 3 N or below, it was confirmed that they could be used in systems such as 3D printing. 

The initial elastic modulus was measured as an indicator to quantify the deformation characteristics of the fabricated hydrogel (Figure 4C). Another parameter, the relaxation time, was measured under a 15% strain condition (Figure 4D). A higher initial elastic modulus indicates that the material can withstand higher levels of stress, thus serving as an indicator of material stiffness. The relaxation time represents the rate at which the material recovers in a deformed state, and a shorter relaxation time indicates higher elasticity of the material [33].

In this study, the fabricated hydrogels exhibited initial elastic modulus values of 20.44 ± 0.93 kPa, 19.36 ± 1.59 kPa, and 16.41 ± 1.34 kPa for HSF0/GG, HSF60/GG, and HSF180/GG, respectively. It was observed that a higher initial elastic modulus corresponded to a shorter relaxation time (Figure 4D). This tendency can be attributed to the formation of a robust hydrogen-bonded double-helix structure between water and the GG backbone, which hinders or restricts the molecular mobility [34]. According to a research finding, a decrease in the length of alginate polymer in ionically crosslinked alginate hydrogels was observed with an increase in the concentration of the ionic crosslinking agent, leading to a faster relaxation time [35]. In conclusion, these mechanical properties of composite hydrogels are influenced more by factors such as hydrogen bonding rather than the molecular weight control of SF by hydrolysis. One reason for the weakened physical crosslinking due to the disassembled SF chain formation could be that the HSF180/GG hydrogel had the lowest initial elastic modulus value.

### 2.4. In Vitro Analysis

#### 2.4.1. Live/Dead Staining 

In the context of tissue engineering, the scaffold plays a crucial role in creating an environment that promotes optimal cell growth and differentiation. To evaluate the biocompatibility and cell attachment capability of the hydrogel, human bone marrow-derived mesenchymal stem cells (hBMSCs) were encapsulated and examined. The results, as depicted in Figure 5, demonstrated the staining of live and dead cells using calcein-AM and ethidium homodimer, respectively, after seven days of culture within the hydrogel. It was observed that majority of cells displayed favorable attachment and viability within all hydrogel groups, with only a small proportion of cells exhibiting signs of apoptosis.

However, upon comparing the fluorescence intensity of dead cells among the different hydrogels, it was noted that the hydrogels incorporating hydrolyzed SF (HSF60/GG and HSF180/GG) exhibited a slightly higher proportion of dead cells in comparison to the HSF0/GG hydrogel (Figure 5B). It is widely recognized that cell attachment is influenced by the stiffness of the hydrogel [36]. The enzymatic degradation of SF within the hydrogel resulted in a reduction in hydrogel stiffness, thereby compromising the development of the F-actin structure in hBMSCs when compared to HSF0/GG hydrogel [37]. Furthermore, a comparison was made between the HSF60/GG and HSF180/GG hydrogels in terms of the fluorescence intensity of dead cells. Interestingly, the HSF180/GG hydrogel displayed a relatively lower fluorescence intensity. This observation suggests that the increased hydrophilicity by hydrolysis of the HSF180/GG hydrogel may have contributed to an enhanced cell attachment capability as compared to the HSF60/GG hydrogel.

#### 2.4.2. Cell-Laden Hydrogel Morphology

It is known that the mechanical properties of hydrogels can influence the cell morphology, which in turn can affect cell differentiation and migration. Therefore, after seeding cells within the scaffolds, the morphology of the cells was examined by bio-Low Vaccum SEM, as shown in Figure 6. The fabricated scaffolds all exhibited a porous structure with randomly oriented features. In this randomly oriented scaffold, the seeded cells typically adopted a rounded shape [38,39]. Across all experimental groups, the cells exhibited a spherical morphology and were uniformly dispersed.

## 3. Materials and Methods

### 3.1. Preparation of Hydrolyzed Silk Fibroin

The HSF aqueous solutions were prepared by the established method [19]. Bombyx mori cocoons (Kyebong Farm, Cheongyang, Republic of Korea) were cut into pieces and 10 g of silk was boiled in a 0.02 M sodium carbonate (Na_2_CO_3_, Showa Chemical, Tokyo, Japan) aqueous solution at 100 °C for 30 min to remove sericin. The degummed silk was washed with distilled water (DW) 3 times and dried in a 60 °C oven to completely remove water. Then, 7 g of the degummed silk was dissolved in 9.3 M of lithium bromide monohydrate (LiBr, Kanto Chemical, Tokyo, Japan) aqueous solution and stored at 60 °C in an oven for 4 h to obtain the silk fibroin (SF) aqueous solution. To hydrolyze SF, 6 mL of a 0.6 M sodium hydroxide (NaOH, Showa Chemical, Japan) aqueous solution was added to the premade SF solution at 80 °C using the drop-wise method. 

To control SF molecular weight, the hydrolysis time was adjusted to 0, 60, and 180 min, referred to as HSF0, HSF60, and HSF180, respectively. HSF0 was used as a control group as a pure silk fibroin aqueous solution. At the specific time points, 3.5 mL of a 1 M hydrochloric acid solution (HCl, Samchun, Pyeongtaek, Republic of Korea) was added in the hydrolyzed SF solution to obtain a pH of 7. Then, the HSF solutions was dialyzed in DW using a SnakeSkin^®^ dialysis tubing (MWCO: 3500, Thermo Fisher Scientific, Waltham, MA, USA) for 72 h. After dialysis, HSF solutions were centrifuged at 9000 rpm, 4 °C, for 20 min, 2 times. The concentration of the final SF solutions was 5–7%. The products were stored at 4 °C until further use. 

### 3.2. Sodium Dodecyl Sulfate-Polyacrylamide Gel Electrophoresis (SDS-PAGE)

The molecular weight distribution (MWD) of the HSF was determined by sodium dodecyl sulfate-polyacrylamide gel electrophoresis (SDS-PAGE of proteins). The concentration of the stacking gel was 5% and the concentration of the separating gel was 15%. A molecular marker of 10–180 kDa (Thermo Fisher Scientific, Waltham, MA, USA) was used for examining the HSF molecular weight. Protein 5X sample buffer (Elpis Biotech, Daejeon, Republic of Korea) was mixed with 2% HSF aqueous solution to make the final 1X loading buffer. Then, 10 μL of each sample was loaded on 15% gel and separated by a Protein Electrophoresis System (Pharm Tek, Seoul, Republic of Korea) at 110 V for 50 min. After electrophoresis, the gel was washed with DW for 5 min and stained with SimplyBlue™ Safe Stain (Invitrogen, Carlsbad, CA, USA) for 3 h. Then, it was washed with DW for 30 min.

### 3.3. Fabrication of Hydrogels

Low-acyl gellan gum (GG; GelzanTM CM, Sigma-Aldrich, Burlington, MA, USA) was dissolved in DW with an amount of 1% (*w*/*v*) and thoroughly stirred for 2 h at 90 °C. The temperature was lowered to 60 °C and 0.06% (*w*/*v*) of calcium chloride (CaCl_2_, Sigma-Aldrich, Burlington, MA, USA) was added and homogeneously mixed for 15 min. Then, 3% each of the prepared HSF0, HSF60, and HSF180 solutions was included in the GG solution. The final concentration of hydrogels contained 0.5% GG, 1.5% HSF, and 0.03% CaCl_2_. The fabricated hydrogels were signified according to the hydrolysis time, as HSF0/GG, HSF60/GG, and HSF180/GG. The prepared hydrogel solutions were poured into a petri dish and solidified at RT for 10 min. The cylindrical-shaped hydrogel samples were prepared by punching with a biopsy punch (Kai Medical Biopsy Punch, Gifu, Japan). Finally, a hydrogel with a diameter of 6 mm and a height of 3 mm was manufactured for further characterization.

### 3.4. Physicochemical Property

#### 3.4.1. Morphological Structure Study

The porous structure was characterized by a bio-LV scanning electron microscope (SEM, Japan, HITACHI, Tokyo, Japan). The fabricated samples were sequentially stored at 4 °C, −20 °C, and −60 °C overnight, respectively. The samples were prepared by cross-sectioning the surface and gold-sputtering for analysis. The pore size distribution was calculated using the ImageJ software V.1.8.0.

#### 3.4.2. Fourier-Transform Infrared Spectroscopy (FT-IR)

Measurements were conducted on hydrogels that had been frozen overnight at 80 °C and lyophilized. Using a Fourier-transform infrared spectrometer (PerkinElmer, Boston, MA, USA) measuring the attenuated total reflectance and Fourier-transform, the composition variation of the fabricated hydrogels was observed in the range of 400 to 4000 cm^−1^. 

#### 3.4.3. Sol Fraction (%)

The fabricated hydrogel samples were lyophilized, and the weight of the dried samples was measured. The samples were then immersed in DW and agitated at 60 rpm for 1 h and lyophilized for 24 h. The sol fraction was calculated by Equation (1): (1)$$Sol\;fraction\;(\%)=\frac{{m_{i}-m}_{f}}{m_{i}}\times 100\;(\%)$$
where *m_i_* is the weight of the initial state of the dried samples and *m_f_* is the weight of the agitated samples.

#### 3.4.4. Swelling Ratio

The prepared samples were immersed in PBS and stored at 37 °C in an incubator for 24 h. The weight of the wet hydrogels was recorded, and the samples were lyophilized. The dry weight of the samples was then recorded. The swelling ratio was calculated by Equation (2):(2)$$Swelling\;ratio=\frac{W_{s}}{W_{d}}$$
where *W_s_* is the weight of the swollen hydrogels and *W_d_* is the weight of the dry hydrogels.

#### 3.4.5. Weight Loss (%)

The weight loss (%) of the hydrogels was analyzed over time for 28 days. The initial weight of the hydrogels was recorded. The samples were immersed in PBS and stored at 37 °C in an incubator. At specific time points, the weight of the hydrogels was measured. The weight loss (%) of the hydrogels was calculated by Equation (3):(3)$$Weight\;loss\;(\%)=\frac{{w_{i}-w}_{s}}{w_{i}}\times 100\;(\%)$$
where *W_i_* is the weight of the initial state of the hydrogels and *W_s_* is the weight of the dry hydrogels at specific time points.

### 3.5. Mechanical Property Analysis

#### 3.5.1. Viscosity Test

The viscosity and the gelation temperature of the hydrogels were evaluated with a viscometer (AMETEK Brookfield, Middleboro, MA, USA). The water circulation bath (VCB-07, JONGRO Industrial Co., Ltd., Seoul, Republic of Korea) was set to 40 °C. After the circulation bath was preheated, the prepared hydrogel solution was transferred to the viscometer (4 mL). The temperature was gradually lowered until it reached 20 °C. The router speed was set at 1 rpm and a cone and plate spindle (SC-32 spindle, AMETEK Brookfield, WI, USA) was used for this study. After gelation was reached, the temperature was heated to 37 °C. The rate of the spindle was changed to analyze the viscosity of the hydrogels depending on the shear rate. 

#### 3.5.2. Injectability Test

The prepared hydrogel solutions were aspirated in 1 mL syringes (Kovax-syringe, Korea Vaccine Co., Ltd., Seoul, Republic of Korea). The syringes were capped with a 26G 1/2″ needle and stored at 37 °C for 5 min. The samples were placed on a custom-designed bracket and the injection force test was carried out with a Texture Analyzer at a speed of 20 mm/min and a load cell of 20 N. The needle was submerged in 37 °C PBS solution to create a physiological environment.

#### 3.5.3. Compression and Relaxation Tests

The compression test was carried out under an unconfined condition. The hydrogels were stored in PBS at 37 °C overnight. The residual solution was removed with filter paper. The height and width of the hydrogels were recorded with a caliper (Mitutoyo, Seoul, Republic of Korea). The compression test was performed with a Texture Analyzer (FTC, Sterting, VA, USA) at the speed of 2 mm/min and a load cell of 10 N. The initial elastic modulus was measured as the slope of the stress–strain curve between 5% and 15% strain. The relaxation test was performed by applying 15% strain and a load cell of 10 N. The stress change over time was measured for 3 min. The stress relaxation time was analyzed by quantifying the time in which the stress of the gel was relaxed to half of the initial stress [40].

### 3.6. In Vitro Study

#### 3.6.1. Cell Culture and Preparation of Cell-Laden Hydrogels

StemPro^®^ BM Mesenchymal Stem Cells (hBMSCs, a15652, ThermoFisher Scientific, Waltham, MA, USA) were applied in this study. Cells were cultured in StemProTM MSC SFM XenoFree medium, which was supplemented with 10% FBS and 1% PS under cell culture conditions (37 °C in 5% CO_2_). The cells were sub-cultured until the passage of the cells reached 6. All the materials were autoclaved before the experiment. The hydrogel solution was prepared under a clean bench. The prepared solution was filtered through a 0.45 µm pore size filter for sterilization. The cells were trypsinized and mixed with the prepared hydrogel solution at 37 °C at an amount of 2 × 10^6^ cells/mL. The cell-laden hydrogels were poured into a petri dish and stored at RT for 10 min to solidify the hydrogels. The cell-laden hydrogels were punched with a 6 mm biopsy punch to obtain samples of 2 mm in height and 6 mm in diameter. The manufactured cell-laden hydrogels were transferred to a 24-well plate and 1 mL of cell culture media was added. The cell culture media was changed every 3 days until further study.

#### 3.6.2. Live/Dead Staining

A live/dead assay was conducted to evaluate the cell viability and cytotoxicity. The hBMSC-encapsulated samples were incubated under standard culture conditions (5% CO_2_ and 37 °C) for 7 days. A live/dead cell-imaging kit (Invitrogen, CA, USA) was used, following the kit protocol. Briefly, the degree of the middle height of the cylindrical hydrogel was cut, and then the cross-sections were transferred to a confocal dish (Cover glass-bottom dish, SPL Life Science, Pocheon, Republic of Korea) and stained with calcein-AM and ethidium homodimer and incubated in the cell culture incubator. The images were obtained by using a Super-Resolution Confocal Laser Scanning Microscope (LSM 880 with Airyscan, Carl Zeiss, Germany) with the process of Z-stack. All measured thicknesses were unified to 200 μm and the fluorescence intensity of live/dead staining was characterized by Image J software V.1.8.0.

### 3.7. Statistics

All the numerical results were presented as the mean ± standard deviation (SD). The GraphPad Prism 5.0 software (GraphPad Software, La Jolla, CA, USA) was utilized to perform the statistical analysis. The studies were analyzed employing a one-way analysis of variance (one-way ANOVA), and the differences were considered significant at *p* < 0.05 (*), *p* < 0.01 (**), and *p* < 0.001 (***).

## 4. Conclusions

In this study, we aimed to modulate the mechanical properties of the dual-biomaterial HSF/GG hydrogel, which holds promise as a potential candidate for tissue engineering, by controlling the hydrolysis of SF. While previous research has focused on adjusting the properties of SF through factors such as the concentration and pH, we sought to simply modify the properties by reducing the length of the SF polymer chains through the process of hydrolysis. Additionally, to create a cell-friendly scaffold, we utilized GG, a biocompatible biomaterial capable of physical crosslinking through ion exchange facilitated by Ca^2+^.

Through evaluation of the mechanical properties and in vitro experiments, we confirmed that SF hydrolysis could enhance the hydrophilicity of SF by removing hydrophobic residues, thereby influencing the physicochemical characteristics of the scaffold, such as pore size enlargement by forming smaller clusters. Moreover, we observed that SF hydrolysis had minimal impact on cell viability, indicating that the HSF/GG hydrogel serves as an environmentally friendly and cell-friendly scaffold without the need for chemical crosslinking agents. 

## Figures and Tables

**Figure 1 molecules-28-05222-f001:**
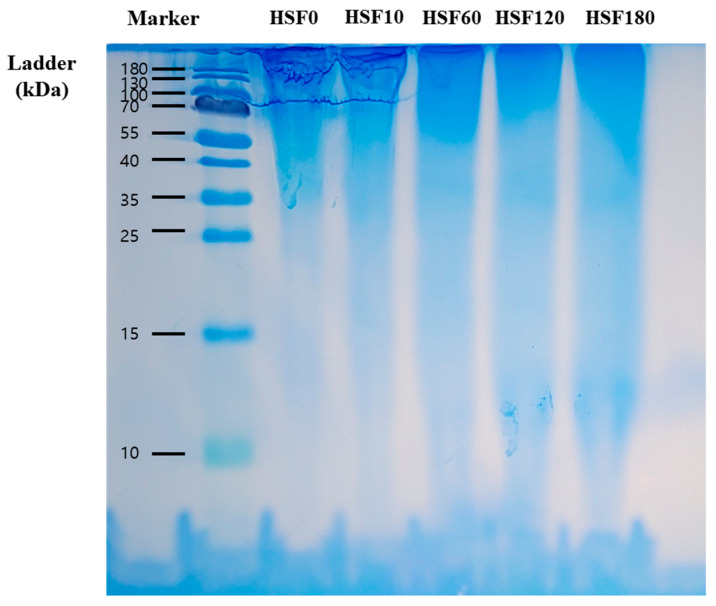
SDS-PAGE of SF solutions. Numbers after HSF indicate the time of hydrolysis, respectively.

**Figure 2 molecules-28-05222-f002:**
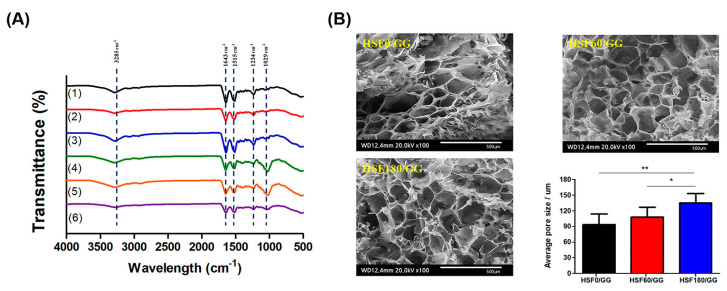
(**A**) FT-IR evaluation of: (1) HSF0, (2) HSF60, (3) HSF180, (4) HSF0/GG, (5) HSF60/GG, and (6) HSF180/GG. (**B**) Morphological observation of the hydrogels and the pore size of the hydrogels (scale bar = 500 μm). Values are mean ± SD, n = 3; *p* < 0.05 (*) and *p* < 0.01 (**).

**Figure 3 molecules-28-05222-f003:**
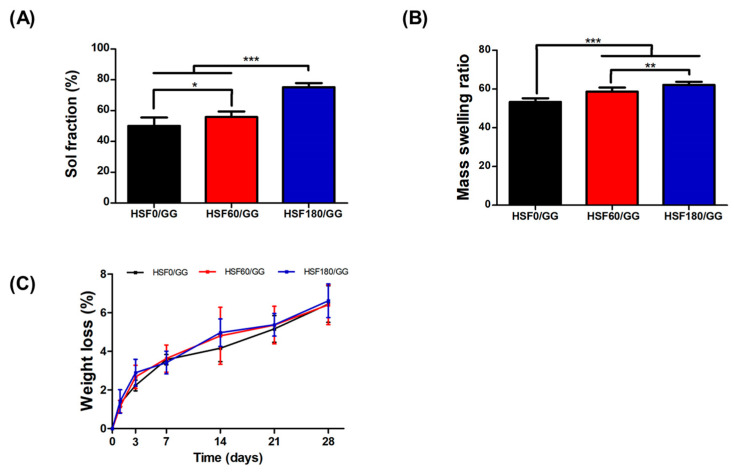
Physicochemical study of the hydrogels: (**A**) Sol fraction (%), (**B**) mass swelling ratio (%), and (**C**) degradation ratio observed for 28 days (%). Values are mean ± SD, n = 9; *p* < 0.05 (*), *p* < 0.01 (**), and *p* < 0.001 (***).

**Figure 4 molecules-28-05222-f004:**
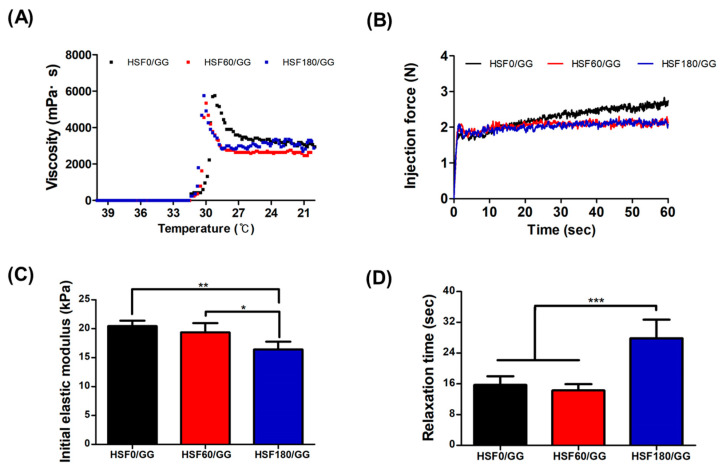
Mechanical characterization of the hydrogels: (**A**) Viscosity test in the temperature range from 40 °C to 20 °C. (**B**) Injection force analysis using a 1 mL syringe at a speed of 20 mm/min and a load cell of 20 N. (**C**) Evaluation of initial elastic modulus measured as the slope of the stress–strain curve between 5% and 15% strain. (**D**) Relaxation test performed by applying 15% strain and a load cell of 10 N (values are mean ± SD (n = 5); *p* < 0.05 (*) *p* < 0.01 (**) and *p* < 0.001 (***)).

**Figure 5 molecules-28-05222-f005:**
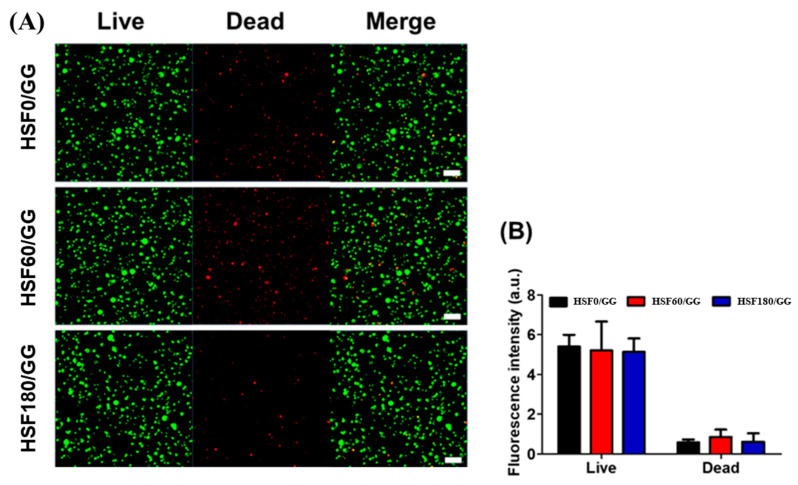
Cell viability test: (**A**) live/dead cell viability assay of hBMSCs cultured in GG/SF hydrogels (scale bar = 50 μm), and (**B**) fluorescence intensity quantification of the staining by ImageJ software V.1.8.0.

**Figure 6 molecules-28-05222-f006:**
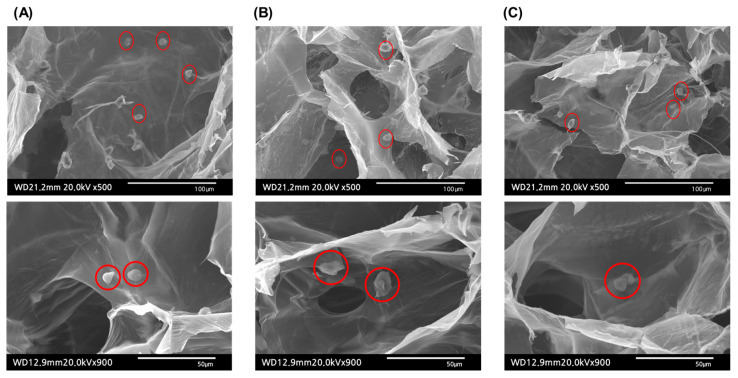
Morphology of hBMSC-encapsulated hydrogels observed under bio-Low Vaccum SEM (cells are represented in red circles): (**A**) HSF0/GG, (**B**) HSF60/GG, and (**C**) HSF180/GG.

## Data Availability

Not applicable.

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
