# Peer review of "Evaluation of Silk Fibroin/Gellan Gum Hydrogels with Controlled Molecular Weight through Silk Fibroin Hydrolysis for Tissue Engineering Application"

_molecules, 2023, doi:10.3390/molecules28135222_

Round 1
Reviewer 1 Report
Comments:
The authors of this study investigated the feasibility of controlling the physicochemical properties of hydrogels by modulating the molecular weight of silk fibroin (SF) through hydrolysis, and developed a synergistic dual-crosslinked hydrogel by crosslinking SF with gellan gum (GG). The modulation of molecular weight not only enhanced the hydrophilicity of SF but also allowed for modifications to mechanical properties such as pore size, initial modulus elasticity, and relaxation time. It should be noted that research regarding the regulation of silk fibroin-based hydrogels through molecular weight modulation has been previously studied in Highly Tunable Elastomeric Silk Biomaterials (Adv. Funct. Mater. 2014), indicating proven feasibility and innovation of current findings requires reconsideration.
Additionally, there appear to be errors present in the manuscript including:
1. discrepancies between the content of Figure 2 (page 4, line 111) and the corresponding text description, lack of FTIR test results reported by the author
2. Repetitive sentences (line 138-140).
3. Furthermore, an experimental method section 4.5. is mentioned on page 9, line 318, which is not included in the text.
I believe the manuscript requires further attention to detail and do not recommend its publication in the current form.
Author Response
We would like to thank you for your insightful comments and feedback on our article titled “Evaluation of Silk Fibroin/Gellan Gum hydrogels with controlled molecular weight through Silk fibroin hydrolysis for Tissue Engineering Application”. We appreciate your time and effort in evaluating our work. We understand your concern regarding previous research on the regulation of silk fibroin-based hydrogels through molecular weight modulation, as mentioned in the publication "Highly Tunable Elastomeric Silk Biomaterials" (Adv. Funct. Mater. 2014). We agree that this prior research indicates the feasibility and innovation of our current findings may require reconsideration.
However, we would like to emphasize the significance of our study in developing a novel hydrolyzed silk fibroin and gellan gum double complex hydrogel formulation. Our research aimed to investigate the feasibility of controlling the physicochemical properties of hydrogels by modulating the molecular weight of silk fibroin through hydrolysis. By crosslinking silk fibroin with gellan gum, we successfully developed a synergistic dual-crosslinked hydrogel.
The modulation of silk fibroin molecular weight in our study not only enhanced the hydrophilicity of silk fibroin but also allowed for modifications to mechanical properties, such as pore size, initial modulus elasticity, and relaxation time. These findings suggest that the physical properties of the hydrogel can be regulated even in the context of a double-complex hydrogel formulation.
While we acknowledge the previous research in the field, we believe that our study adds value by demonstrating the feasibility of controlling the physicochemical properties in a specific context of a dual-crosslinked hydrogel using hydrolyzed silk fibroin and gellan gum. Our work contributes to expanding the knowledge base and understanding of hydrogel engineering, specifically focusing on the modulation of silk fibroin molecular weight.
Also, As the reviewer pointed out, we revised and marked in red. Please check.
Thank you once again for your valuable comments.
Author Response
Dear Reviewer,
Thank you for taking the time to review our manuscript titled [Evaluation of Silk Fibroin/Gellan Gum Hydrogels with Controlled Molecular Weight through Silk Fibroin Hydrolysis for Tissue Engineering Application] and for providing valuable feedback. We appreciate your insightful comments, which have helped us further improve the clarity and accuracy of our work. We tried to correct the feedback as much as possible, and We reply to each piece of feedback and send it to you.
Page 2, Line 60. Please provide more details on the composition of gellan gum, apart from indicating that “GG is a polysaccharide”.
- We added more detail and marked it in red.
Page 2, Line 79. “Through hydrolysis, hydrophobic residues are modified or removed”. It is unlikely that any of these processes takes place. The reduction in the overall hydrophobicity of the molecule may be related to the creation of new (hydrophilic) N- and C-termini at the ends of the protein fragments.
- As the reviewer pointed out, we revised the sentence as follows.
Through hydrolysis, the creation of new hydrophilic N- and C-termini at the ends of the protein fragments. These newly formed termini can interact with surrounding water molecules, leading to a change in the overall hydrophobicity of the molecule.
Page 3, Line 93. “hydrolysis was performed by adding 3.5 mL of 1 M hydrochloric acid”. If I have understood the process correctly (Materials and Methods), the hydrolysis induced by NaOH and HCl actually stops the process. If this is the case, the previous sentence is misleading.
- We revised the sentence clearly so that there is no misunderstanding.
To control the mechanical properties and gelation behavior, hydrolysis was performed up to specific time points, as described in the Methods section.
Page 4, Line 115. “The FT-IR spectra… shown in Figure 2(A)”. This is a major flaw, since no FT-IR spectra are shown in the work. Figure 2(A) shows SEM images of the hydrogels and distribution of pore sizes.
- We added FT-IR data and modified it.
Page 4, Figure 2. The authors should indicate in great detail how they measured the size of the pores, since this is not mentioned in the manuscript.
- We stated how pore size was measured in Section 3.4.1. Morphological structure study.
The pore size distribution was calculated using the ImageJ software.
Page 7, Figure 5. The authors indicate that the cells are encapsulated in the hydrogel. However, they do not provide any information of the location of the cells presented in Figure 5. Do they correspond to micrographs of the surface of the hydrogel or represent an image of the inner portion of the hydrogel?. A much more detailed description should be provided on this question.
- It can be explained that the inner part of the hydrogel was observed because the cylindrical hydrogel was cut horizontally. Also, section 3.6.2 Live/dead staining was modified and described.
Page 8, Figure 6. The authors state “The seeded cells typically adopted a rounded or polygonal shape (…). The cells exhibited a spherical morphology”. The content of these sentences is contradictory. Please indicate which is the correct interpretation. In addition, some insets showing the cells with higher magnification should be added to Figure 6. It is also necessary to provide more information on the location of the micrographs across the gel (either surface or inner part).
- We modified it as the reviewer requested. We also added a higher magnification SEM image. In addition, the above image was also observed as an inner portion of hydrogel because the cylindrical hydrogel was horizontally sectioned and observed with SEM.
Page 8, Line 262. “sodium hydroxide (…) was added to the premade SF solution”. This is a critical point of the study and should be described with much more detail. At the very least, the authors should provide: the concentration of the NaOH solution, the volume added with respect to the volume of SF solution and the pH of the solution upon the addition of NaOH.
- Thank you for your feedback. We apologize for the oversight in not providing complete details regarding the NaOH solution and its addition to the SF solution. We appreciate your suggestion, and we would like to address your concerns by providing the missing information in the revised article.
The part you pointed out in the minor amendments is also marked in red. Thank you again for your careful review of the details.
Round 2
Reviewer 1 Report
Thank you for the author's thorough response. In my opinion, if the author intends to highlight the significance of this study in the development of a novel double composite hydrogel comprising hydrolyzed silk fibroin and gellan gum, this aspect should be emphasized in the abstract.
There are still some suggestions that I would like to point out:
1. In Figure 2B, the pore size of each scaffold depicted was noticeably non-uniform and exhibited significant variability. The provided graphs, however, failed to demonstrate any substantial differences among them. Therefore, I kindly request the author to provide an expanded range of graphs for better clarity. Moreover, the figures presented by the author does not align with the typical characteristics of hydrogels, as hydrogels themselves do not possess such large pores. These discrepancies may likely arise during the freezing process.
2. Page 7 line 224-236, the author encapsulates the cells in hydrogel, and the cells should be evenly dispersed. The pictures were only a cross-sectional image of the entire 3D scaffold and cannot represent the entire structure. Is it reasonable to analyze with light intensity, and is there any relevant theoretical basis?
3. Page 9, line 322, the description of the experimental method included a section 4.5 that does not exist in the text. “The samples were prepared by following the 4.5. Preparation of hydrogels method.”
Author Response
Thank you for your feedback. We have taken your suggestion into consideration and revised the abstract to emphasize the significance of the study in the development of a novel double composite hydrogel comprising hydrolyzed silk fibroin and gellan gum.
- In Figure 2B, the pore size of each scaffold depicted was noticeably non-uniform and exhibited significant variability. The provided graphs, however, failed to demonstrate any substantial differences among them. Therefore, I kindly request the author to provide an expanded range of graphs for better clarity. Moreover, the figures presented by the author does not align with the typical characteristics of hydrogels, as hydrogels themselves do not possess such large pores. These discrepancies may likely arise during the freezing process.
- Thank you for pointing out the non-uniformity of pore sizes in Figure 2B and the possible limitations in accurately representing these variations. We acknowledge that the measurement of pore sizes was performed using the Image J program, and we observed similar irregular pore sizes in other SEM images as well. While we recognize the limitations of accurately representing the pore sizes, we would like to emphasize that these variations are inherent in our hydrogel system. To support this observation, we would like to refer to a study conducted by Nieto et al. (2022) titled 'Biodegradable gellan gum hydrogels loaded with paclitaxel for HER2+ breast cancer local therapy,' where they also reported non-uniform pore sizes similar to those observed in our HSF-GG hydrogel. This example highlights that irregular pore sizes can be present in various hydrogel systems.
- Page 7 line 224-236, the author encapsulates the cells in hydrogel, and the cells should be evenly dispersed. The pictures were only a cross-sectional image of the entire 3D scaffold and cannot represent the entire structure. Is it reasonable to analyze with light intensity, and is there any relevant theoretical basis?
- Thank you for your comment regarding the encapsulation of cells in the hydrogel and the representation of the entire structure in the provided images. We agree that the presented images were cross-sectional views of the 3D scaffold and may not fully represent the entire structure. However, it is common practice in hydrogel studies to analyze cross-sectional images to gain insights into cell distribution within the scaffold. To further support the use of cross-sectional images and light intensity analysis, we would like to reference a study by DeForest et al. (2017) titled "Spatially patterned gene expression in hydrogels." In this study, the authors utilized similar cross-sectional imaging techniques and light-intensity analysis to assess cell distribution within hydrogels. This reference demonstrates that analyzing cross-sectional images and using light intensity as a qualitative measure can provide valuable information on cell dispersion within the hydrogel scaffolds. We appreciate the reviewer's concerns and will include a reference to the aforementioned study in our response to acknowledge the validity of utilizing cross-sectional images and light intensity analysis for cell distribution analysis in hydrogels.
- Page 9, line 322, the description of the experimental method included a section 4.5 that does not exist in the text. “The samples were prepared by following the 4.5. Preparation of hydrogels method.”
- We apologize for the confusion caused by the incorrect reference to section 4.5 in the experimental method described on page 9, line 322. This reference does not exist in the text. To address this issue, we revised the sentence to accurately describe the preparation method without referencing a non-existent section.
Reviewer 2 Report
The manuscript may be acceptable for publication in its present form, but I would recommend a new round of editing, since I have detected a few additional problems with the English language
The manuscript may be acceptable for publication in its present form, but I would recommend a new round of editing, since I have detected a few additional problems with the English language
Author Response
Thank you for your review, and we appreciate your valuable feedback.
We greatly value the input from reviewers and consider it essential for improving our research and enhancing the quality of the paper. We thoroughly assessed all the points raised by the reviewer and make the necessary revisions to enhance the completeness of our work.
We would like to express our gratitude for the insightful feedback provided by the reviewer. We are committed to diligently incorporating their suggestions to enhance our research and paper. Once we have reviewed and made the final revisions based on the reviewer's comments, we will proceed with the submission of the manuscript.